# Systematic single cell RNA sequencing analysis reveals unique transcriptional regulatory networks of *Atoh1*-mediated hair cell conversion in adult mouse cochleae

**Shu Tu, Jian Zuo** *

Department of Biomedical Sciences, Creighton University School of Medicine, Omaha, NE, United States of America

* jianzuo@creighton.edu

**Data Availability Statement:** Source code available at https://github.com/shutu0302/sc-paper.

## Abstract

Regeneration of mammalian cochlear hair cells (HCs) by modulating molecular pathways or transcription factors is a promising approach to hearing restoration; however, immaturity of the regenerated HCs in vivo remains a major challenge. Here, we analyzed a single cell RNA sequencing (scRNA-seq) dataset during Atoh1-induced supporting cell (SC) to hair cell (HC) conversion in adult mouse cochleae (Yamashita et al. (2018)) using multiple high-throughput sequencing analytical tools (WGCNA, SCENIC, ARACNE, and VIPER). Instead of focusing on differentially expressed genes, we established independent expression modules and confirmed the existence of multiple conversion stages. Gene regulatory network (GRN) analysis uncovered previously unidentified key regulators, including *Nhlh1*, *Lhx3*, *Barhl1* and *Nfia*, that guide converted HC differentiation. Comparison of the late-stage converted HCs with the scRNA-seq data from neonatal mouse cochleae (Kolla et al. (2020)) revealed that they closely resemble postnatal day 1 wild-type OHCs, in contrast to other developmental stages. Using ARACNE and VIPER, we discovered multiple key regulators likely to promote conversion to a more mature OHC-like state, including *Zbtb20*, *Nfia*, *Zmiz1*, *Gm14418*, *Bhlhe40*, *Six2*, *Fosb* and *Klf9*. Our findings provide insights into the regulation of HC regeneration in adult mammalian cochleae in vivo and demonstrate an approach for analyzing GRNs in large scRNA-seq datasets.

## Introduction

The development of sensory hair cells (HCs) in the inner ear is regulated by complex gene regulatory networks (GRNs) involving transcription factors and their associated cofactors, or regulons. Understanding the temporal dynamics of these GRNs is crucial for the study of hearing loss pathology and HC regeneration for hearing restoration [1]. While non-mammalian HCs can regenerate through the trans-differentiation of supporting cells (SCs) to HCs, adult mammalian cochlear HCs do not have this capability [2, 3]. Efforts to regenerate HC-like cells in

**Funding:** 1. JZ; NIHR01DC015010; 2. JZ; NIHR01DC015444; 3. JZ; ONR-N00014-18-1-2507; 4. JZ; USAMRMC-RH170030; 5. JZ; LB692/ Creighton The funders had no role in study design, data collection and analysis, decision to publish, or preparation of the manuscript.

**Competing interests:** The authors have declared that no competing interests exist.

adult mouse cochleae in vivo have had limited success, with the newly converted HCs (cHCs) from SCs remaining immature and non-functional [4, 5]. A key strategy for inducing regeneration of HCs in mammals is the induction of ectopic expression of the transcription factor Atoh1 in SCs, leading to the formation of Atoh1-induced cHCs [6]. However, the extent to which these cHCs resemble endogenous HCs is still debated. Some studies suggest that these *Atoh1*-induced cHCs are similar to endogenous postnatal day 1–7 (P1-7) HCs based on early HC marker expression and electrophysiological properties [7–8]. It also remains debated whether *Atoh1*-induced regeneration in adult mouse cochleae follows the endogenous developmental path of HC differentiation and maturation [9, 10]. Addressing these questions will provide guidance to future efforts to regenerate functional HCs in adult mammalian cochleae.

To gain a deeper understanding of the transcriptional regulation of Atoh1-induced HC regeneration, we performed a comprehensive analysis of the regulatory network involved in the SC-to-HC conversion process. We used a combination of different gene network modeling analyses to identify and characterize the transcription factors and their target genes involved in the conversion. In order to find more potential genes that can promote maturation, we integrated expression data of pro-sensory cells and outer hair cells (OHCs) at postnatal day 1 and 7 from a scRNA-seq dataset of neonatal mice [1] with the dataset of Atoh1-induced cHCs in adult mice [7]. We found that the most differentiated cHCs resemble postnatal day 1 wildtype OHCs when compared with other OHC developmental stages. Moreover, we identified key regulators, including *Zbtb20*, *Nfia*, *Zmiz1*, *Gm14418*, *Bhlhe40*, *Six2*, *Fosb* and *Klf9*, as candidate regulators that may promote the regeneration of more functional HCs in adult mammals. Our study serves as an example of how to perform in-depth analysis with large scRNA-seq datasets and provides important insights into the regulatory dynamics of cochlear HC regeneration, which could guide future experimental approaches to hearing restoration in vivo.

## Results

### Identification of inner hair cells (IHCs), outer hair cells (OHCs) and converted hair cells (cHCs) in adult mouse cochleae through scRNA-seq analysis using Scanpy

The field of scRNA-seq data analysis is dominated by two popular tools, Scanpy [11] and Seurat [12]. Scanpy, a python implementation of scRNA-seq analysis package, was inspired by Seurat [11]. In this study, we employed Scanpy to re-analyze our previous scRNA-seq data from dissociated adult mouse organs of Corti [7]. The data was obtained from the Fgfr3-iC-reER; *Atoh1*-HA; Chrna9-EGFP; tdTomato mouse model. The re-analysis revealed 3734 cochlear cells, comprising of 161 Atoh1-HA+ cHCs, 24 IHCs and 43 OHCs (**Fig 1A and 1B**). We identified cHCs based on their co-expression of Atoh1 and HA genes, driven by SC-specific CreER (*Fgfr3*). As shown in Fig 1B and 1C, using Scanpy, we were able to resolve inner and outer HC populations (IHCs and OHCs) that were previously grouped together in Seurat analysis. Distinct molecular signatures of Atoh1-HA+ cHCs, DCs/PCs, IHCs and OHCs were also confirmed by studying the highly differential genes of these cell types (**Fig 1C and 1D**). Our analysis demonstrates the effectiveness of using Scanpy to identify unique cell type population with low cell counts in intricate scRNA-seq datasets.

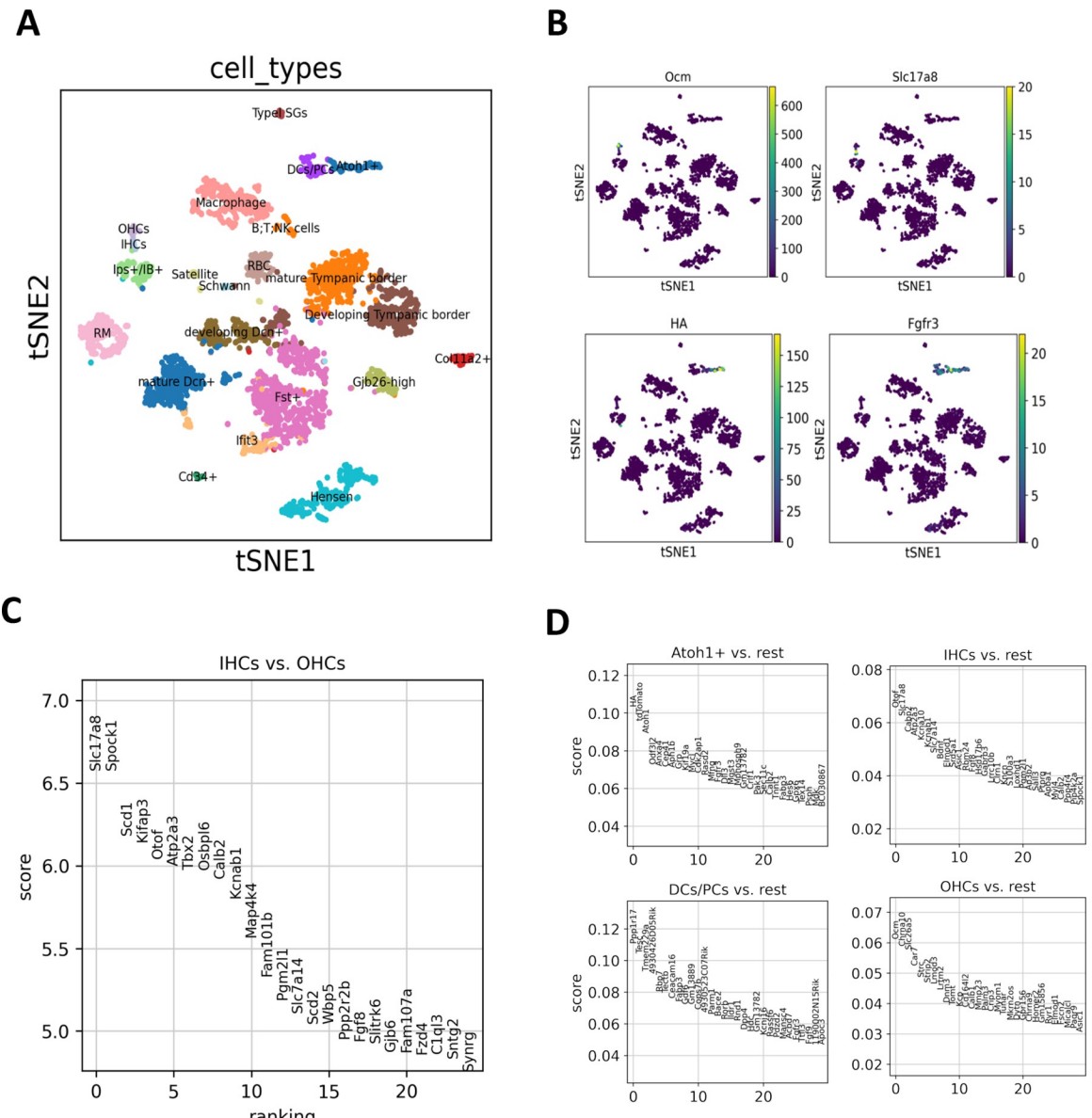

**Fig 1. Single-cell gene expression profiling of adult mouse cochleae during ectopic Atoh1-induced conversion. (A)** tSNE plot of the SCANPY clusters. Cell types were annotated based on previous cell annotations [7]. **(B)** tSNE mapping of normalised count expression levels of specific markers. Slc17a8 and Ocm were highly expressed in mature IHCs and OHCs, respectively, but not in cHCs (annotated as Atoh1$^+$ in the figure) or SCs (annotated as DC/PCs, IPs/IB+ in the figure). HA (Atoh1-HA) and Fgfr3 were highly expressed in cHCs clusters. **(C)** Top 25 differential genes when comparing IHCs with OHCs. **(D)** Ranking for highly differentially expressed genes in the Atoh1-HA+ cHC, DC/PC, IHC and OHC clusters, respectively. RM: Reisner membrane; DCs: Deiters' cells; PCs: pillar cells; HCs: hair cells; Iphs: inner phalangeal cells; IBs: inner border cells; SGs: spiral ganglia.

## Identifying cell type-specific gene regulatory networks in single-cell RNAseq data using SCENIC and WGCNA: Validation and comparison of results

In our previous publication, we identified three progressive stages of cHC maturation: cHC1, cHC2 and cHC3, and demonstrated continuous gene expression change during the conversion from SC to cHC using DEGs [7]. We also identified transcription factors that promote

conversion [7]. However, we did not perform co-expression network analysis or more specific GRN analysis. In this study, we aimed to identify cell type specific GRNs for the cells involved in the conversion process (**Fig 2A**). These GRNs could provide insights into the mechanisms related to a more efficient conversion.

**WGCNA analysis.** Despite its design for bulk RNAseq data, WGCNA can be utilized to identify functional gene clusters and key regulators in single-cell transcriptome data by constructing co-expression networks from gene expression profiles [13–16]. These clusters, called modules, comprise genes that interact and perform similar functions.

We applied WGCNA to the top 4000 variable genes in the SC-cHCs consortium (single cell clusters of SC at P12 and P33, cHC1, cHC2, cHC3), which is a standard selection for single-cell highly variable gene analysis (**S1A Fig**). The resulting six modules included genes that are co-expressed and associated with changes in expression pattern during conversion. The module size varied significantly, with the turquoise module being the largest (1049 genes) and the red module being the smallest (33 genes) (**S1 Table**). By relating module expression to conversion stages, we found that the turquoise module was highly co-expressed with cHC3, the blue module with SCs; the yellow module with cHC1, and the red module with both cHC1 and cHC2 (**Fig 2A**). Moreover, we observed a negative correlation between the blue and turquoise modules, indicating distinctive expression profiles of SCs and cHC3 (**Fig 2B**). The grey modules, containing genes that did not belong to any modules, were excluded from further analysis.

We used the eigengene approach (See Methods) to derive representative expression profiles for the four co-expression modules (blue, red, yellow and turquoise) and then computed expression correlations between genes using the Algorithm for the Reconstruction of Accurate Cellular Networks (ARACNE) [17] (**S2A-S2C Fig**). The ARACNE algorithm allows for the identification of key regulators in each module that can regulate their downstream transcription factors. We identified several key regulators in each module (summarized in **Table 1**), including SC markers in the blue module and regulators of cHC regeneration in the turquoise module. The red and yellow modules were found to represent cHC1-2 with fewer regulators. Functional gene enrichment analysis of gene modules further confirms the correlation of each modules with the corresponding conversion states (**S3A-S3D Fig**).

**SCENIC analysis.** The single cell regulatory network inference and clustering (SCENIC) was developed to analyze GRNs and identify regulons, groups of co-regulated genes in single-cell RNAseq data [18]. To identify and characterize the transcription factors and their target genes involved in the conversion process, we included endogenous HCs from the same study. The SCENIC analysis revealed prominent cell-type specific regulons for SCs, cHCs, OHCs and IHCs, as shown in **Fig 2C**. Highlighted in the blue box, IHCs, SCs, and early cHCs (cHC1) share the expression of specific regulons, such as Creb1, Gata3, Zfp467, Sox13, Sox2, Sox10, Mxi1, Hes1, and Sox9, while intermediate (cHC2) and late (cHC3) stages of cHCs exhibit decreased expression of these regulons. We know from the previous publication that overexpression of *Atoh1* can promote the conversion of SCs to a HC-like state [6, 19, 20]. Based on our findings, cHC1 have similar key regulons to SCs, while cHC3 shows a downregulation of regulons and an upregulation of regulons highlighted in the orange box, which are also active in OHCs. Furthermore, bulk RNA-seq profiles [7] supported the notion that cHCs resembled OHCs at P7 more than the mature SCs and the mature OHCs/IHCs analyzed. Hence, we focused on comparing the transcription profiles of the cHCs with those of the OHCs in subsequent analysis to identify regulators that can promote progression from cHC3 towards OHC-like stages. Comparison of the results from SCENIC and WGCNA revealed shared regulons that were predominantly active in the late stage of conversion, as shown in **Fig 2D**.

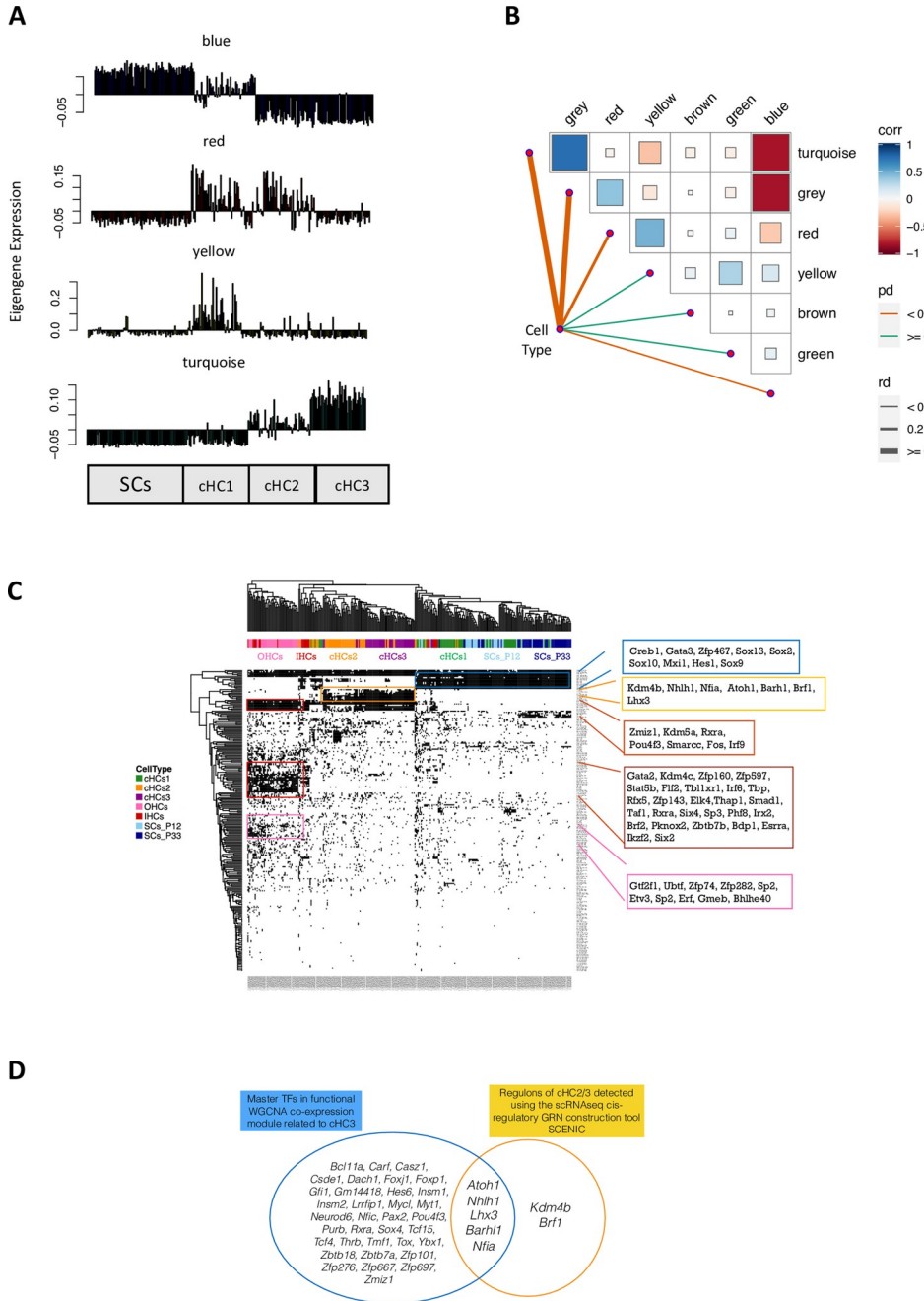

**Fig 2. Gene regulatory network (GRN) inference using WGCNA and SCENIC. (A)** WGCNA analysis of cHCs and endogenous HCs, showing the correlation of each cell type and the expression pattern of eigengenes in each module. Each spike represents one cell of the cell types indicated below. **(B)** Correlation matrix depicting the correlation coefficients of different modules, with all seven modules included and self-comparisons removed from the graph. The size of the box and its color gradient indicate the correlation coefficient. **(C)** SCENIC identified cell-type-specific GRNs during SC-to-cHC conversion. The binary regulon activity matrix shows correlated regulons (absolute correlation > 0.3) active in at least 1% of cells. Cells are colored according to cluster identity, and genes are colored based on expression level. Regulons exceeding a manually adjusted AUC threshold are shown in black, while inactive regulons are white. Each stage of the conversion is shown at the top of the graph in different colours. **(D)** Venn diagram displaying the five overlapping key regulators identified by both WGCNA turquoise module (cHC3) and SCENIC in cHC2/3, compared to regulons in OHCs.

**Table 1. Regulators in WGCNA-derived co-expression modules.**

| Module | Representative cells | Regulators |
|---|---|---|
| Blue | SC | *Camta1, Crem, Emx2, Gata3, Glmp, Hes1, Hey1, Id1, Id2, Id3, Ikzf2, Irx3, Nfe2l3, Rora, Rorb, Sall2, Sox10, Sox9, Zfp36l1, Zfp467* |
| Red | cHC1, cHC2 | *Zbtb20* |
| Yellow | cHC1 | *Hes5, Sox2, Hey1* |
| Turquoise | cHC3 | *Atoh1,* ***Barhl1****, Bcl11a, Carf, Casz1, Csde1, Dach1, Foxj1, Foxp1, Gfi1, Gm14418, Hes6, Insm1, Insm2,* **Lhx3***, Lrrfip1, Mycl, Myt1, Neurod6,* **Nfia***, Nfic,* **Nhlh1***, Pax2, Pou4f3, Purb, Rxra, Sox4, Tcf15, Tcf4, Thrb, Tmf1, Tox, Ybx1, Zbtb18, Zbtb7a, Zfp101, Zfp276, Zfp667, Zfp697, Zmiz1* |

Note: Highlighted transcription factors are also identified using SCENIC in **Fig 2C**.

## Expression pattern of converted HC cluster 3 (cHC3) resembles wild-type P1 OHCs more than other naive OHC stages

In the past, we identified cHC3 as the most differentiated cHCs in P33 cochleae, which had a similar resemblance to differentiating neonatal HCs. The bulk RNA-seq profiles of cHCs at P33, as well as OHCs at P7 or P22, found that P33 cHCs were similar to P7 OHCs [7]. To further investigate this, we re-analyzed raw data from a recent scRNA-seq study covering wild-type cochlear cells across E14, E16, P1 and P7 [1]. Trajectory inference (TI) using the PAGA (Partition-based graph abstraction) method in the dynoverse package [21] was applied to Lateral Pro-sensory Cells (LPsCs), OHCs, Deiters' Cells (DCs), Inner Pillar Cells (IPCs), and Outer Pillar Cells (OPCs) at the four aforementioned ages. Unlike Monocle analysis [1], PAGA TI generated distinct branches from LPsCs (**Fig 3A**). LPsCs committed to HCs follow a trajectory path from LPsCs to iHC_E14 and iOHC_E16, and finally into OHC at P1 and P7. Meanwhile, other lineages of LPsCs give rise to supporting cells like IPC or OPC. Nevertheless, we still observed some uncertainties in the trajectory, such as the backward arrow pointing from OHC_P1 to iOHC_E16. PAGA assumes that the trajectory is a tree-like structure with clear directionality, but some biological processes may not follow a strict linear progression. This can lead to inaccuracies in the analysis of certain cell types or developmental stages.

We combined OHCs at different developmental stages (E14, E16, P1 and P7) [1] with cHCs (cHC1-3) and mature OHC at P33 from the previous study [7]. The two datasets were then anchored using the top variable genes of each cluster (**Fig 3B**). We tested the performance of the tool SCTransform [22] for normalization and found that it generally performed better than the standard log normalization included in Seurat and Scanpy (**Fig 3C**). OHC_P1 formed two subsets as it was prepared in two batches in the original study [1]. The heatmap for the top 10 DEGs in each cluster revealed that cHC3 expressed cell markers of OHC_P1 alongside an additional panel of genes (**Fig 3D**). Genes exhibiting high expression in cHC1-3 but limited expression in wild-type OHC_P1 may be due to persistent ectopic Atoh1 expression, previous SC fates, and/or the requirement for additional transcription factors. Correlation plot of cell types revealed that cHC3 resembled OHC_P1 more (r = 0.32) compared to other naive OHC stages (iHC_E14: r = 0.1; iOHC_E16: r = 0.2; OHC_P7:r = 0.03; OHC_P33: r = 0.03) (**Fig 3E**). We concluded that the ectopic expression of *Atoh1* in DCs and PCs of adult mouse cochleae could convert them into P1-state OHC-like cHCs but failed to further convert them to fully mature OHCs (such as OHCs at P7 and P33).

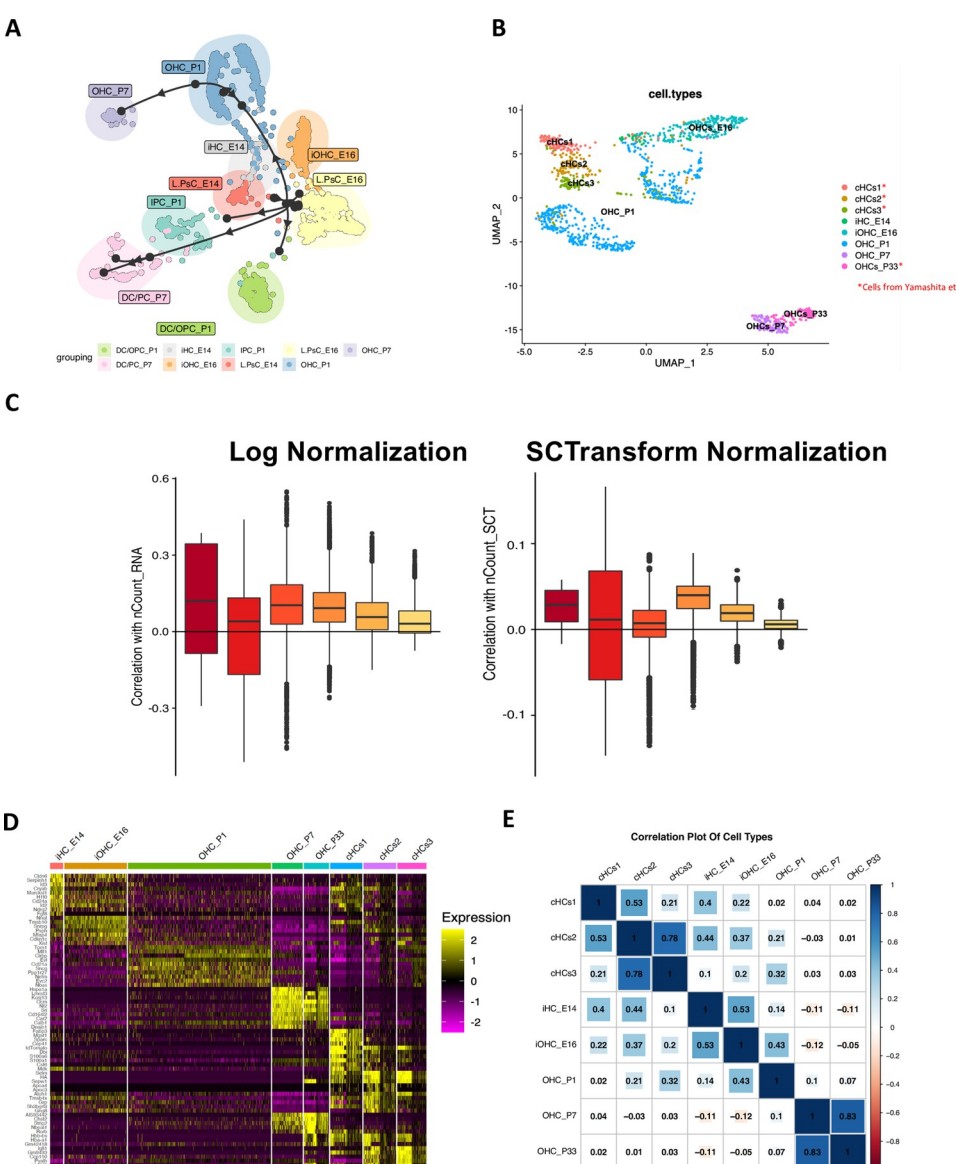

**Fig 3. Comparison of expression patterns of converted HC clusters to P1 wild-type OHCs and other naive OHC stages. (A)** PAGA trajectory analysis of wild-type cochlear cells across different developmental stages. L.PsCs, Lateral pro-sensory cells; IPC: Inner pillar cells; DC/OPCs: Deiters' cells/Outer pillar cells. **(B)** UMAP analysis of eight different cell types, including cHC1-3 and OHC_P33 from Yamashita *et al*. [7], and OHC at different stages of natural hair cell development [1]. iHC_E14: immature hair cell at E14; iOHC_E16: immature outer hair cell at E16. **(C)** The SCT-normalization method effectively reduced the influence of technical factors in the combined dataset. **(D)** Heatmap showing the top 10 (or all, if less than 10) markers for each cluster. **(E)** Correlation plot of cell types revealed that cHC3 is more similar to OHC_P1 (r = 0.32) than to other naive OHC stages (iHC_E14:r = 0.1; iOHC_E16: r = 0.2; OHC_P7: r = 0.032; OHC_P33: r = 0.031).

## Identification of key regulators between cHC3 and endogenous OHCs at P1 and P7

We utilized the ARACNE and VIPER algorithms [23, 24] to identify potential regulators that may facilitate the progression of cHC3 towards endogenous P7 OHCs. In brief, the VIPER (Virtual Inference of Protein-activity by Enriched Regulon analysis) algorithm performs an enrichment statistical analysis on every regulon, using the regulatory network provided by

ARACNE and the expression matrix, to identify the most significant transcription factors associated with the regulatory models derived from the comparison of different cell types.

Fig 4 displays the top 10 regulators identified by VIPER in the cHC3: OHC_P7 [1, 7] comparative analysis. *Atoh1* was identified as the most significant candidate, which was expected given the ectopic *Atoh1* expression in cHCs. The right-side color panels show the expression value and estimated protein activity of each transcription factor in the dataset. These parameters indicate the significance of these genes in shaping the characteristics of either cHC3 or OHC_P7. Compared with OHC at P7, cHC3 exhibited significantly higher activity of regulons regulated by *Atoh1, Zbtb20, Nfia, Zmiz1, Gm14418 and Prox1*. Among these six gene up-regulators, only *Prox1* was deemed to be insignificant when using 100 bootstraps in the analysis, while the four down-regulators (*Bhlhe40, Six2, Fosb and Klf9*) were found to be significant in both single and 100 bootstraps (S4 Fig). *Atoh1* has the largest regulon with 272 genes, followed by *Zbtb20* with 179 genes and *Nfia* with 121 genes. Among the down-regulators, *Bhlhe40* and *Six2* had the largest number of associated genes with 58 and 42, respectively.

## Discussion

In this study, we overcame the challenges of analyzing scRNA-seq GRNs by integrating multiple scRNA-seq datasets and utilizing multiple recently developed analytical tools. Our new analysis approach focuses on the connections among co-expressed genes to establish independent expression modules, which is superior to previous analyses that only focused on differentially expressed genes. Integrating scRNA-seq data from multiple platforms can be challenging due to differences in library preparation, sequencing depth, and experimental conditions, leading to batch effects that can obscure biological signal and confound downstream analysis. We found SCTransform can provide improved expression correlation results. We then applied multiple GRN analysis tools and revealed continuous regulatory program dynamics for Atoh1-induced SC-to-HC conversion with multiple GRNs and regulons. These findings provide insights into the process of SC-to-HC conversion and will guide future experimental approaches to achieving more efficient conversion both in cochlear explants and in vivo [7, 25, 26].

Through comparison of the transcriptome profiles in our SC-to-cHC regeneration in adult mouse cochleae and the LPsC-to-OHC transcriptomic changes in neonatal mouse cochleae [1], we found that the most differentiated cHC stage in adult mouse cochleae resemble P1 OHCs more closely than other naive OHC stages such as P7 (Fig 3E). This finding can be explained by the constitutive ectopic expression of *Atoh1* in SCs during the conversion, which is crucially regulated during natural HC development. Normally, *Atoh1* expression increases from E13-14 to P0-1 and then drastically decreases between P1 and P7 [20]. However, in the cHCs of our study, *Atoh1* overexpression is driven by Cre-mediated constitutive CAG promoter, which continuously drives *Atoh1* ectopic expression in adult SCs and cHCs at a persistent and high level without decreasing. Therefore, our data suggest that the proper duration of *Atoh1* ectopic expression in SCs is required for further differentiation and maturation of cHCs. A recent study has shown that transient ectopic expression of Atoh1 can drive SC-to-HC conversion more effectively than constitutive ectopic expression of Atoh1 [27].

Moreover, our analysis revealed cHCs have induction of additional genes that are not expressed during natural HC development, likely due to persistent *Atoh1* ectopic expression, previous SC fates, and/or the need for additional transcription factors. These findings suggest the necessity of manipulating additional transcription factors or regulons concurrently or consecutively to ensure appropriate and efficient conversion.

In this regard, multiple key regulators, including *Atoh1, Barhl1, Lhx3, Nfia, Nhlh1*, were identified independently by both WGCNA and SCENIC as potential key players in guiding

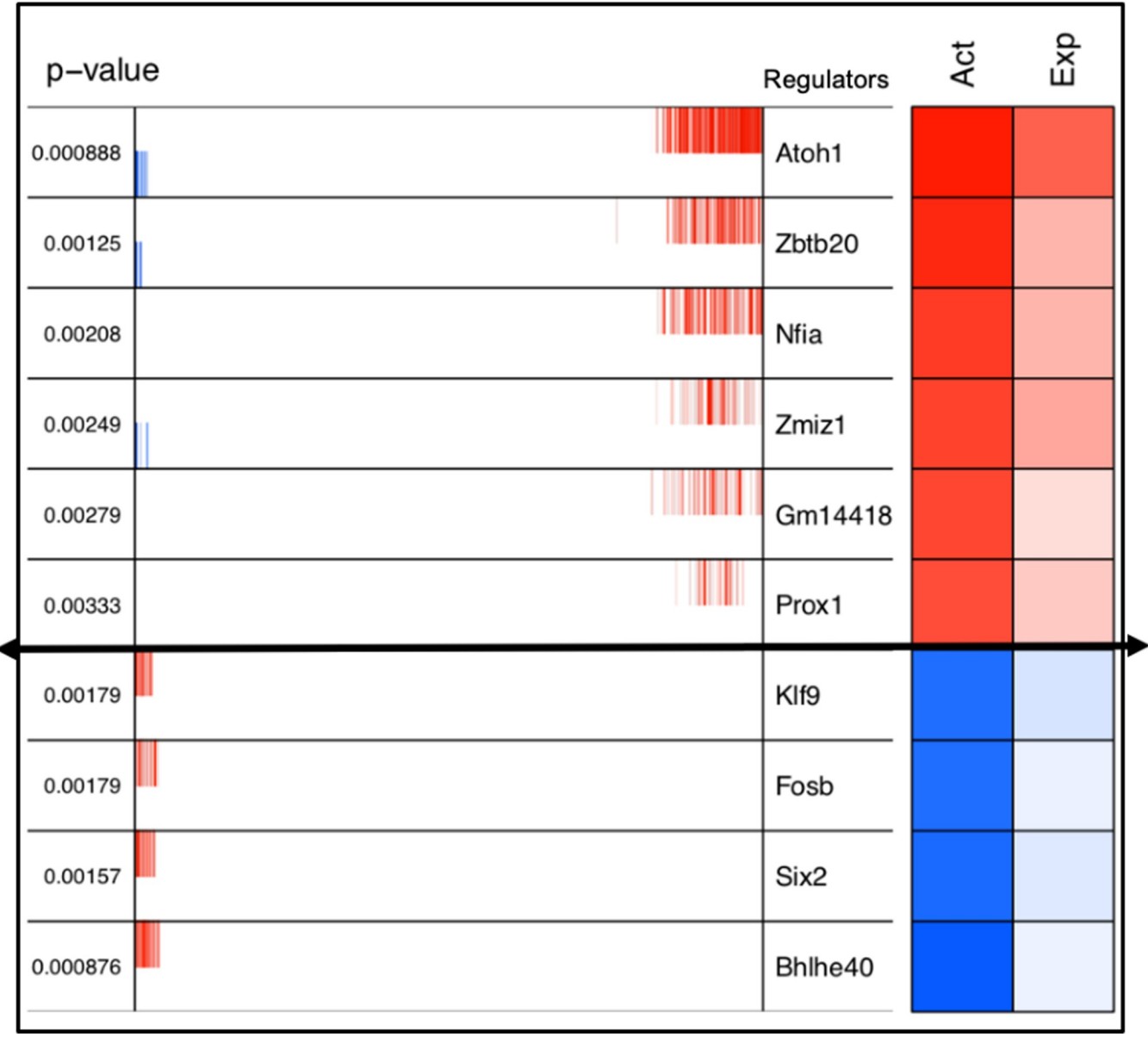

## Top-10 TFs (Regulators) found comparing cHCs3-P7OHC (VIPER plot)

**Fig 4. Transcription factors identified using VIPER in the comparison of the expression profiles of cHC3 versus naive OHCs at P1.** VIPER plot shows the top 10 up-regulated (red) and down-regulated (blue) transcription factors. The strength of protein activity (Act) and RNA expression (Exp) are represented by color intensity, with darker color indicating higher values. Prox1 is shown to be insignificant when using 100 bootstraps. The lines in the middle of the plot are the individual gene expression values for each cell in the dataset and provide additional information about the variability in gene expression within each cell type.

cHC1-3 differentiation. *Barhl1* and *Lhx3* are known to be essential for HC development and differentiation, with Atoh1 maintaining expression of Barhl1 [20, 28, 29]. In the developing inner ear, *Barhl1* expression appears after *Atoh1* initial expression and formation of hair cells

from otic progenitors [30]. *Nhlh1* has also been considered a "pro-hair cell gene" that is necessary for the formation and maintenance of hair cells [31].

In addition to these regulators, our analyses using ARACNE and VIPER algorithm identified additional key regulators, including *Bhlhe40*, *Klf9*, *Fosb*, *Nfia*, *Six2*, *Zbtb20* and *Zmiz1*, which are likely to promote cHCs-to-OHC_P7 conversion. Most of these regulators have not been previously studied for SC-to-HC conversion.

Together, our re-analyses of the two datasets [1, 7] have provided additional insights into the regulators that could promote SC-to-HC conversion in adult mouse cochleae, offering guidance for further studies on OHC maturation. Experimental validation of our bioinformatic results is beyond the scope of this study. However, integrating scRNA-seq datasets obtained from non-mammalian vertebrates, such as zebrafish and chicken HC regeneration, could provide further insights into the gene regulatory network underlying the progression of supporting cells to differentiated hair cells.

## Methods

### 10x genomics single-cell RNA-seq data processing

FASTQ files from the Gene Expression Omnibus data repository under GSE173217 were processed by the function "cellranger count" of CellRanger (10× Genomics, Version 6.0.2) to generate feature counts of every single cell. The reference genome was the mouse genome GRCm38 (10× Genomics, Version 5.0.0). To remove the cells with low quality, cells with gene number over 200, and the ratio of mitochondria lower than 15 were maintained, and genes with at least one feature count in more than three cells were used for the following analysis. The bam file from GSE137299 was processed and converted to FASTQ and the same CellRanger analysis was applied to process the data.

### Single-cell RNA-seq analysis

The obtained gene-barcode matrices were further analyzed using Scanpy [11] or Seurat 4.0.1 [12] in R. Cells with relatively small and large library size in individual datasets were individually removed as potential doublets and low-quality cells. Cell cycle classification was performed using the Cell-Cycle Scoring and Regression method published in Seurat, and cells that were classified as being in G1 phase were chosen. Differentially expressed genes were identified using the FindAllMarkers function in the packages with a default condition and PCA followed by tSNE and UMAP analysis were further performed using the differentially expressed genes. The two datasets were then anchored using the top variable genes of each cluster to minimize the batch effect. SCTransform was then utilized to normalize and integrate the datasets. Top10 markers were used to plot the heatmap and the correlation plot. Corrplot package (v0.88) was used for the correlation plot (https://github.com/taiyun/corrplot).

### Pseudotime analysis

Trajectory inference was conducted using the dynoverse R packages (https://github.com/dynverse/dyno) [21]. Raw count and expression matrices were extracted from the different cell clusters and wrapped into a dynwrap object using dynwrap::wrap_expression. Grouping and dimensionality reduction information were added to the dynwrap object using add_grouping and add_diregulatored, respectively. We selected TI methods using the guidelines_shiny tool for our specific dataset, and the methods retained for analysis were slingshot, PAGA-tree and multiple spanning tree (MST). While every method provided specific insight into the data structure, PAGA-tree best recapitulated the structure of other methods combined in our data.

We obtained pseudotime visualization using calculate_pseudotime on the PAGA-tree model object. Branches in the cell trajectory represent cells that have alternative gene expression patterns.

## WGCNA analysis

The WGCNA (v1.70) package [32] was used in R to identify co-expression gene modules in the cell types involved in the ectopic Atoh1-induced HC conversion. After filtering genes, gene expression values were imported into WGCNA to construct co-expression modules using the automatic network construction with default settings. To identify any obvious outliers and visualize how the samples are clustered, we then constructed the sample dendrogram to visualize how the samples are clustered and to identify any obvious outliers (**S1A Fig**). All samples were included in the clusters and passed the cut-off thresholds.

The power value is a critical parameter that can affect the independence and average connectivity degree of the co-expression modules. Therefore, network topology using different soft thresholding powers was screened, and $\beta = 4$ (scale free $R^2 = 0.9$) was selected for further analysis (**S1B Fig**). We then constructed a matrix of adjacencies by calculating Pearson correlations between all pairs of genes across all selected samples. The matrix was then computed into a Topological Overlap Matrix (TOM) using the function TOMsimilarity [33]. The TOM, referred to the interconnection between two genes, was used as input for hierarchical clustering analysis, and a cluster of genes with high topological overlap was defined as a module.

Finally, we identified modules and constructed the GO analysis of the four modules correlated with them. The results were plotted using the R package clusterprofile [34] (**S3A-S3D Fig**), and significant enrichment was considered at $p < 0.05$.

## ARACNE analysis using eigengenes from WGCNA modules

We next sought to derive the eigengenes and their GRNs from WGCNA modules. Eigengenes were calculated using the moduleEigengenes function to identify the first principal component of each gene module as a representative. We used the Algorithm for the Reconstruction of Accurate Cellular Networks (ARACNE) [17] to compute expression correlations between genes in the four co-expression modules (blue, red, yellow and turquoise).

To account for the highly variable and noisy single-cell gene expression profiles and the dropout effect of mRNA, we used the PISCES package (https://github.com/califano-lab/PISCES). The ARACNE algorithm [17] was then employed to compute the regulatory interactions, avoiding many indirect interactions that are generally found through those co-expression methods that are less accurate. We filtered the MI values to select only those corresponding to the regulatory events between transcription factors (considered regulators), and their linked genes (i.e., the targeted genes, considered as regulons).

The networks of the transcription factor module with motif information were visualized by Cytoscape [35].

## SCENIC analysis

To compare with the GRNs derived from WGCNA analysis, we used SCENIC [18] to investigate GRNs for the SCs, cHCs and HCs in the scRNA-seq data published by Yamashita *et al.* [7]. To filter out genes that were most likely noise, we used the geneFiltering function in the SCENIC package and retained genes that were detected in at least 1% of the cells. Additionally, only the genes available in the RcisTarget databases were kept. SCENIC identifies potential GRNs through the following steps: (1) identifying transcription factors and candidate target genes based on co-expression using GENIE3, (2) constructing regulons by filtering modules

for candidate genes enriched in their transcription factor binding motifs utilizing RcisTarget, with mm10 transcription factors used as the reference, (3) determining the activity of each regulon in each cell, and (4) constructing a regulon activity matrix to cluster cells based on shared regulatory networks. Through this approach, we identified cell states based on shared activity of a regulatory subnetwork. Finally, we used the default "AUCCellThreshholds" for each regulon as the threshold to binarize the regulon activity scores and created the "Binary regulon activity matrix".

## VIPER analysis

To identify key regulators that promote regeneration, we compared the transcriptomes of cHC3 and OHCs at P1 and P7 using the algorithm called VIPER (Virtual Inference of Protein-activity by Enriched Regulon analysis) (v1.22) [24]. First, we used the ARACNE algorithm to reverse-engineer context-specific regulatory networks by leveraging a large collection of gene expression profiles. Then, we inferred the activity of each regulator by computing the enrichment of the genes in its regulon (from ARACNE) in each cell type. Positive or negative enrichment indicates up- or down-regulation of the regulator, respectively, and its activity is represented by the normalized enrichment score.

We used PISCES to build metacell matrices, and included bootstrapping in the VIPER analysis, testing 100 times subsets of the cells to identify the most stable regulators. We selected the most significant transcription factors with a p-value $< 0.05$.

We found five significantly overexpressed transcription factors and four significantly repressed transcription factors when comparing cHC3 and P7OHC. To test whether the difference between P1OHC and cHC3 affected the results, we also compared P1OHC+cHC3 with P7OHC. These comparisons indicated the most significant difference of the transcriptomic regulator related to the different HC or HC-like groups (cHC3-P7OHC, P1OHC-P7OHC).

## Supporting information

**S1 Fig. WGCNA analysis in cHCs and endogenous HCs. (A)** Sample clustering was conducted to detect outliers. This analysis was based on the expression data of top 4000 variable genes in the total 251 cells with 161 Atoh1-HA+ cHCs and 90 SCs available in Yamashita *et al* [7] (S2 Table). Genes were clustered based on a dissimilarity measure. The branches correspond to modules of highly interconnected groups of genes. Colors in the horizontal bar represent the five cell types assigned by scRNA-seq analysis. **(B)** Selection of the soft-thresholding powers for scale-free co-expression network. The left panel showed the scale-free fit index versus soft-thresholding power. The right panel displayed the mean connectivity versus soft-thresholding power. Power 4 was used.
(ZIP)

**S2 Fig. Gene co-regulation network for the four co-expression modules. (A-D)** The network was derived using the ARACNE program and constructed using Cytoscape. The key regulators (in red) are connected with their targets (in purple).
(ZIP)

**S3 Fig. Functional analysis of gene modules associated with hair cell conversion. (A)** Gene ontology (GO) analysis of genes in the blue module (SC—cHC1) showed enrichment of biological processes related to early-stage hair cell development, such as Notch signaling pathway, regulation of mechanoreceptor differentiation, and epithelial proliferation **(B)** GO analysis of the red module (cHC1—cHC2) revealed pathways related to organ growth and morphogenesis, cell and tissue migration, indicating the dynamic regulation of conversion-associated

genes in this middle stage. **(C)** GO analysis of the yellow module (cHC1—cHC2) showed enrichment of genes associated with similar processes as the red module, supporting their potential role in the middle stage of hair cell conversion. **(D)** GO analysis of the turquoise module (cHC2 –cHC3) suggested pathways such as synapse organization and neuroepithelial cell differentiation, indicating their potential role in the later stages of hair cell conversion. (ZIP)

**S4 Fig. VIPER analysis identified the TFs active in OHC_P7 and cHCs.** Table showing the top 10 up- and down-regulated transcription factors found by VIPER with corresponding parameters (normalized enrichment score (NES), p-value, false discovery rate (FDR), and bootstraps). (TIF)

**S1 Table.** (CSV)

**S2 Table.** (XLSX)

## Acknowledgments

We thank the Zuo lab members, Dr. Yusi Fu, Dr. Jun Xia, Dr. Ming Lei and Ms. Alexander Grassam-Rowe for comments.

## Author Contributions

**Conceptualization:** Shu Tu.

**Data curation:** Shu Tu.

**Formal analysis:** Shu Tu.

**Funding acquisition:** Jian Zuo.

**Investigation:** Shu Tu.

**Methodology:** Shu Tu, Jian Zuo.

**Resources:** Jian Zuo.

**Software:** Shu Tu, Jian Zuo.

**Supervision:** Jian Zuo.

**Validation:** Shu Tu, Jian Zuo.

**Visualization:** Shu Tu, Jian Zuo.

**Writing – original draft:** Shu Tu, Jian Zuo.

**Writing – review & editing:** Shu Tu, Jian Zuo.

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
