## [Decision Letter · Decision Letter 0]

8 Sep 2022

PONE-D-22-20296Systematic single cell RNA sequencing analysis reveals unique transcriptional regulatory networks of Atoh1-mediated hair cell conversion in adult mouse cochleaePLOS ONE

Dear Dr. Zuo,

Thank you for submitting your manuscript to PLOS ONE. After careful consideration, we feel that it has merit but does not fully meet PLOS ONE’s publication criteria as it currently stands. Therefore, we invite you to submit a revised version of the manuscript that addresses the points raised during the review process. *As you can see, our reviewers were enthusiastic about the need for such a manuscript in the inner ear field, but they do express some reasonable reservations that can be addressed. In addition, please note that the data and code sharing should be well annotated and easily followed by someone with a bioinformatics background. In its current state (https://github.com/shutu0302/sc-paper-code), the shared code is not well annotated and appears piecemeal. As inspiration, here's an example of well annotated and shared code that can be easily reproduced (Tosches et al 2018, Science. https://github.molgen.mpg.de/MPIBR/ReptilePallium/).*

We look forward to receiving your revised manuscript.

Kind regards,

Taha A Jan, MD

Academic Editor

PLOS ONE

Journal Requirements:

2.We note that the grant information you provided in the ‘Funding Information’ and ‘Financial Disclosure’ sections do not match. 

"We thank the Zuo lab members, Dr. Yusi Fu, Dr. Jun Xia, Dr. Ming Lei and Ms. Alexander Grassam-Rowe for comments. This work was supported in part by NIHR01DC015010, NIHR01DC015444, ONR-N00014-18-1-2507, USAMRMC-RH170030, and LB692/Creighton."

"1. JZ; NIHR01DC015010; 

2. JZ; NIHR01DC015444;

3. JZ; ONR-N00014-18-1-2507;

4. JZ; USAMRMC-RH170030;

5. JZ; LB692/Creighton

Reviewers' comments:

Reviewer's Responses to Questions

**Comments to the Author**

1. Is the manuscript technically sound, and do the data support the conclusions?

Reviewer #1: Partly

Reviewer #2: Partly

2. Has the statistical analysis been performed appropriately and rigorously? 

Reviewer #1: No

Reviewer #2: Yes

3. Have the authors made all data underlying the findings in their manuscript fully available?

Reviewer #1: Yes

Reviewer #2: Yes

4. Is the manuscript presented in an intelligible fashion and written in standard English?

Reviewer #1: Yes

Reviewer #2: No

5. Review Comments to the Author

Reviewer #1: This study conducts timely re-analysis of previously published scRNAseq datasets of the mouse cochlea, attempting to move past the relatively simpler analysis of differential gene expression and focusing on gene networks and searching for ‘master regulator’ transcription factors. As such, the approach is interesting and of use for the community. It is also an example of expanding the usefulness of scRNAseq datasets beyond initial publications and the added value of combining datasets to address new questions.

However, there are some concerning oversights in the data analysis workflow. As one of the stated goals of the paper was to provide an example of integrated analysis of readily available scRNAseq datasets, the authors should provide a lot more detail about how this was achieved. Moreover, it is not clear from the data analysis as presented if the authors managed to properly integrate the two datasets to identify and correct for the typical ‘batch effect’ confounding factors. I do think that there is some indication within the data for a gene expression profile linking cHC3s to P1-OHCs. But this should be re-evaluated under a much stricter and clearer dataset integration workflow.

More specific comments follow:

Lines 110-116. Please provide more details as to why you expect ‘more refined clusters’ when using knn graphs on scanpy, keeping in mind PlosONE is a general audience journal.

Figure 1. First, the number of Atoh1-HA+ cHCs and HCs identified here is different to the ones reported in the original paper. Comments?

You are arguing that the clustering algorithm used this time around is better. How so?

You are identifying separate clusters for IHCs and OHCs. Have you tested cluster stability? Could you identify known OHC vs IHC genes differentially expressed between these clusters?

For the WGCNA analysis, I’m confused as to which cells were used for this analysis. The figure legend for fig S1 says 228 SCs and cHCs were used. But, as informed in the results section, 228 cells is the exact sum of 161 Atoh1-HA+ cHCs + 24 IHCs + 43 OHCs. How many SCs are there in the dataset?

Why was this analysis only run on SCs and cHCs? Why not include HCs?

Supplementary figure 1C is important for the interpretation of the WGCNA analysis. It should be moved to the main manuscript and properly annotated. What’s the information conveyed by the size of the squares? Cell types? pd? rd?

Why did you use 4000 variable genes for WGCNA? You are working on 10x data; so, how many genes were detected per cell? Using these many variable genes can risk looking into detection threshold noise, and not really into differential gene co-expression between cell types, especially considering the very low number of cells used in this analysis. Do you identify the same, or similar, modules using a smaller number of variable genes?

How were the eigengenes in figure 2A computed?

The ARACNE analysis is interesting and seems to be providing some new insights. Could you provide more details as to how master regulator genes are designated?

Which/how many variable genes were used for the SCENIC analysis?

Figure 2C compares ‘regulons’ identified with WGCNA with those from SCENIC. However, these analyses were run in different subsets of cells (no HCs were included in WGCNA). Could you then provide a rationale for the comparison?

‘Converted HC cluster 1-3 (cHC1-3) expression patterns resemble P1 wild-type OHCs development with some additional genes expressed’ subtitle. Consider rephrasing. What’s the meaning of ‘some additional genes’?

The methods section describing the PAGA trajectory inference analysis is missing.

The PAGA trajectory analysis is quite central to the stated objectives of the paper. As such, it should be described in greater details and the relevant figures moved to the main manuscript.

In figure S4A, I fail to see the ‘dynamic process underlying multi-lineage LPsC differentiation’. Could you provide further explanation?

There is virtually no information about how the integration of the OHCs (E14, E16, P1, P7) with the cHCs was performed. This is a crucial step that affects all the data analysis and conclusions moving forward. Step by step details on the workflow should be provided in the methods section.

I’m assuming the P33 OHCs were from Yamashita, et al. Why use only those and not the data from other ages from that publication?... it is very confusing to follow which cells were included in which analysis throughout the paper, even when carefully looking at the original publications. This should be more clearly stated throughout the results, figure legends and methods sections.

I understand you are testing the hypothesis that cHCs may show features of (immature) OHCs. But, why run the integrated analysis only with OHCs? Why not include IHCs to see if they are indeed more distant? And why not run the integration with SCs also? Having common cell types can greatly improve anchoring during integration. Have you considered integrating the full two datasets?

Figure 3A shows a UMAP representation of the pooled samples. Without more information of how the two datasets were integrated, the interpretation of this (and all subsequent) plots is weakened.

Did you run the integration on Seurat/scTransform? Was the clustering then run on the residuals?

Lines 220-221: ‘Specifically, a small group of cHC3 are clustered together with OHC_P1…’ it’s unclear from this and from the plot in figure 3A if you are talking about re-clustering after integrating the two datasets, or if you mean that a few cHC3 cells land, on the UMAP plot, on the OHC_P1 region.

If the former, how was the clustering done? If the latter, what’s different about those cHC3 cells?

Lines 222-227 are confusing and make it even less clear how the data integration was run. Also, were there uncorrected batch effects within the original datasets?

Heatmaps of DEGs do not validate UMAPs. How were the top10 DEGs computed? There are hemoglobin and mitochondrial genes in the heatmap, raising concerns that what we are looking at is greatly influenced by batch differences between the two datasets, as opposed to accurately describing the differences and similarities between OHCs and cHCs.

At points it seems like the authors confuse clustering with dimensionality reduction/visualisation.

Lines 230-233. ‘Heatmap of the top 10 DEGs for each cluster clearly showed that during the dynamic change from cHC1-3, a panel of genes were expressed, and this resembled the natural HC development from E14 to P1 (Figure 3B).’ This is almost discernible on the heatmap (top right corner), but completely overrun by the expression of genes in the bottom right. Again, data integration workflow and batch correction need to be properly addressed and described to support the above statement. Also, to discuss dynamic changes trajectories should be compared, not clusters.

Why are not P33-OHCs included in figure 3C?

The trajectories in figures S4B-C seem to be more affected by the differences between datasets than by maturation. Again, a clearer explanation on how the datasets were integrated is needed to properly run and interpret this trajectory inference analysis.

Line 258. There are 10 genes, not 20, in figure 4A

Figure 4. Atoh1 is overexpressed on cHC3 cells, so it would be expected to show up as such. It should be clarified in the text.

And further explanation of the interpretation of figures 4A-B would be useful. Is Prox1 significant or not?

What’s the rationale behind the umap plot in figure 4C?

Lines 307-310. The conclusions stated are not fully supported by the data analysis as it stands. Admittedly, there are some indications that similar expression patterns may be followed by normal HC development and during HC conversion. But, it is not clear from the data analysis how the two datasets were integrated and how the trajectory inference were done. Moreover, to reach conclusions as to the similarity or differences between trajectories, the relevant branches need to be directly compared, an analysis that was not performed here.

Line 318. What do the authors mean by “aberrant paths”?

Reviewer #2: In this manuscript, the authors re-analyze published single-cell RNA-sequencing data using new bioinformatic tools and draw new insights regarding SC-to-HC conversion via Atoh1 overexpression in the mouse cochlea. scRNA-seq data can harbor rich information that may not always be fully unmasked in a single study. As new analytical tools become available, these data can be further analyzed to reveal new details, an approach that could prove very helpful in understanding the complex biology, for example, of cellular regeneration. Overall, the authors make a good pitch for why performing this kind of work might be helpful and present insights/results that could guide future work on the important topic of efficiently converting SCs to replace lost HCs.

Below I list several points that the authors should address to bring the manuscript closer to a publication-ready state. At a broad level, my suggestions relate to two main issues: 1) the authors should think more carefully about the relevance of their data to claims they make. In several instances, their conclusions are not necessarily supported by the data they cite, either because restrictions regarding data interpretability were ignored, or the data as presented doesn’t seem to be directly related. 2) the writing needs to be improved, most importantly, to avoid statements that are either too broad or misleading.

SPECIFIC POINTS

1. The term ‘master regulator’ appears to be used quite loosely throughout the paper and is hence misleading. For example, in lines 333-336, the authors use this terminology to refer to several differentially expressed transcription factors, whose functions in SC-to-HC fate conversion have not even been studied to date (by their own admission). What makes them ‘master regulators’ as opposed to just ‘transcriptional regulators’? In addition, in lines 92-93, the authors state that master regulators “…are defined as transcription factors that differentially regulate sets of target genes (regulons).” This is not a correct definition of MRs. The authors are describing a regulatory molecule, not a master regulator. Here’s an example of a study that discusses and makes recommendations on this topic: https://www.ncbi.nlm.nih.gov/pmc/articles/PMC3718559/

The authors should instead use ‘regulators’, or ‘transcriptional regulators’ or ‘transcription factors’ where applicable throughout their manuscript.

2. It is well known that distance in the UMAP space doesn’t necessarily indicate transcriptional similarity. The authors even acknowledge this point explicitly in lines 227-229. Yet, they make the claim that cHC3 and OHC_P1 have similar transcriptomes because of cluster proximity in lines 219-222. This should be corrected. The authors are however not wrong in suggesting that cHC3s appear more similar to OHC_P1 than older OHCs; support for this is clearer in the correlation matrix in Figure 3C and the heatmap in Figure 3B.

3. Correction necessary in lines 229-230: A heatmap (of top DEGs) doesn’t validate a UMAP.

4. In lines 237-240, the authors suggest that genes that were highly expressed in cHC1-3 but showed limited expression in OHC_P1 could be downstream targets of Atoh1. This explanation doesn’t make sense: Why would Atoh1-regulated targets be selectively expressed in cHC1-3 when OHC_P1 also express Atoh1?

5. In lines 243-245, the authors suggest that their results in Figures S4B-S4C suggest that maturation stages of cHC1-3 are “similar to the natural HC development”. Those plots do show trajectories across different cell stages/types but cannot be used to make inferences regarding similarity of developmental progression. The heatmap in Figure 3B does provide some support to this statement though, so the authors are advised to reconsider the way they frame this point.

6. Panels in Figure 4 are highly redundant. Most if not all of Figure 4B should be supplemental; anything remaining could be merged with Figure 4A. For example, size of the regulon could be useful in evaluating the data in Figure 4A, so those values could be directly indicated in Figure 4A and the rest of Figure 4B could be moved to a Supplementary Figure.

7. It’s unclear how data in Figure 4B, as cited, support the claim made in lines 309-310.

8. In line 320, the authors state “These findings further solidify Atoh1-mediated regenerative approaches for hearing restoration…”. It is not clear at all how any of their results allows such a claim.

9. Although the authors have mostly done a decent job of describing the rationale behind the work and their results, there is clearly some room for improvement in writing. This ranges from typos (e.g., line 319), possible mistakes (e.g., “insignificant” in line 266) to unsupported claims (e.g., “…acting as master regulators…” in lines 90-91 where the wording should’ve been “…may act as…”), unintelligible framing (e.g. “…Based on VIPER, the with the top 20 MRs found in the cHC3: OHC_P7 comparative analysis was presented in Figure 4A.” in line 258) and inappropriate content structure (e.g., lines 258-273 contain text written almost like figure legend, not results).

6. PLOS authors have the option to publish the peer review history of their article (what does this mean?). If published, this will include your full peer review and any attached files.

Reviewer #1: No

Reviewer #2: No

---

## [Author Response · Author response to Decision Letter 0]

26 Dec 2022

Response to reviewer 1

1. Lines 110-116. Please provide more details as to why you expect ‘more refined clusters’ when using knn graphs on scanpy, keeping in mind PlosONE is a general audience journal.

Answer: We provided detailed explanation in that paragraph: “In the construction of kNN graph, each cell is a node with edges to its k nearest neighbors. SNN is built on the basis of kNN, but two cells are only connected by an edge if they share a nearest neighbor in the kNN graph. Neighbors of each cell in the kNN graph are ranked from 1 (the same cell, because each cell is its own closest neighbor) to k (the most distant neighbor). Edges in the SNN graph are weighted according to the best of the average ranks of their shared neighbors. Therefore, kNN construction is usually used to identify a cell type while SNN is used to eliminate false positives and false negatives.”

2. Figure 1. First, the number of Atoh1-HA+ cHCs and HCs identified here is different to the ones reported in the original paper. Comments?

Answer: In the original paper, Seurat 1.4.0.16 is used for the analysis and they used scran 1.4.5 to do the cell cycle classification to choose cells that were classified as being in G1 phase. Here, when classifying cells using cell cycle, we used the Cell-Cycle Scoring and Regression method published in Seurat 4.0.1. Therefore, the number of cells (G1 phase) qualified will be different as the markers for different cell cycle phases and regression methods are different.

You are arguing that the clustering algorithm used this time around is better. How so?

Answer: Our analysis with Scanpy can resolve inner and outer HC (IHC and OHC) populations. These two populations form a single cluster from previous analysis with Seurat using the same clustering parameters. 

You are identifying separate clusters for IHCs and OHCs. Have you tested cluster stability? Could you identify known OHC vs IHC genes differentially expressed between these clusters?

Answer: We added a Figure 1C to show the ranking for the highly differential genes in Atoh1-HA+ cHCs, DCs/PCs, IHCs and OHCs, respectively. It’s clear to see IHCs and OHCs cluster express their own markers. 

For the WGCNA analysis, I’m confused as to which cells were used for this analysis. The figure legend for fig S1 says 228 SCs and cHCs were used. But, as informed in the results section, 228 cells is the exact sum of 161 Atoh1-HA+ cHCs + 24 IHCs + 43 OHCs. How many SCs are there in the dataset?

Answer: Thank you so much for correcting this mistake. WGCNA used SCs and cHCs (cHC1, cHC2 and cHC3). Total cell number is 251, with 161 A161 Atoh1-HA+ cHCs and 90 SCs. We attached the cell expression matrix in a supplementary.

Why was this analysis only run on SCs and cHCs? Why not include HCs?

Answer: We did not include HCs because the HCs in this dataset are all all mature cells and have a very distinct expression profile than SCs and cHCs. Mixing with SCs and cHCs will confound the results as there is no expected development link between cHCs and HCs. The aim of WGCNA analysis was to reveal the gene expression profile changes throughout the SCs to cHCs.

 Supplementary figure 1C is important for the interpretation of the WGCNA analysis. It should be moved to the main manuscript and properly annotated. What’s the information conveyed by the size of the squares? Cell types? pd? rd?

Answer: We movded figure 1C to the main manuscript and added expanations in the main text. The size of the squares indicated corr. 

“We found the modules of blue, red and yellow are positively correlated with each other, indicating the similarity of their profiles, while yellow and turquoise are negatively correlated (Figure 2B). The negative correlation might result from the relative distinctive profile of cHC1 and cHC3.” 

Why did you use 4000 variable genes for WGCNA? You are working on 10x data; so, how many genes were detected per cell? Using these many variable genes can risk looking into detection threshold noise, and not really into differential gene co-expression between cell types, especially considering the very low number of cells used in this analysis. Do you identify the same, or similar, modules using a smaller number of variable genes? 

Answer: We have tried using 1000, 2000, 4000 and 6000 top variable genes to construct WGCNA. We found using top 4000 DEGs we can still get clear distinct profiles from the four cells types as shown in Figure 2A. When using less number of genes, we obtain less modules and cannot distinguish different types of cHCs.

How were the eigengenes in figure 2A computed?

Answer: The eigengenes were calculated using moduleEigengenes function in the WGCNA package.

The ARACNE analysis is interesting and seems to be providing some new insights. Could you provide more details as to how master regulator genes are designated?

Answer: We added in the text: “ARACNe is a novel algorithm, using microarray expression profiles, that is specifically designed to scale up to the complexity of regulatory networks in mammalian cells, yet general enough to address a wider range of network deconvolution problems19.”

 Which/how many variable genes were used for the SCENIC analysis?

Answer: The whole gene expression profiles were used at the beginning of the analysis. Then we used the standard geneFiltering function in the SCENIC package to filter genes that are most likely noise and kept genes that are detected in at least 1% of the cells. Finally, only the genes that are available in RcisTarget databases will be kept.

We updated this in the method.

 Figure 2C compares ‘regulons’ identified with WGCNA with those from SCENIC. However, these analyses were run in different subsets of cells (no HCs were included in WGCNA). Could you then provide a rationale for the comparison?

Answer: The aim of WGCNA analysis was to reveal the gene expression profile changes throughout the SCs to cHCs. We did not include HCs because the HCs in this dataset are all all mature cells and have a very distinct expression profile than SCs and cHCs. Mixing with SCs and cHCs will confound the results as there is no expected development link between cHCs and HCs. However, the advantage of SCENIC is that it is designed for scRNAseq and it. There has not been any study that performed and compare both WGCNA and SCENIC. Therefore, we expected there would be difference between the WGCNA and SCENIC. It is also inspiring that we could found 5 same gene regulators, suggesting their possible roles in the conversion process.

 ‘Converted HC cluster 1-3 (cHC1-3) expression patterns resemble P1 wild-type OHCs development with some additional genes expressed’ subtitle. Consider rephrasing. What’s the meaning of ‘some additional genes’? Answer: ‘some additional genes’ refers to the genes that are highlt expressed in cHCs while not expressed in P1 navie OHCs. Shown in Figure 3a.

The methods section describing the PAGA trajectory inference analysis is missing. The PAGA trajectory analysis is quite central to the stated objectives of the paper. As such, it should be described in greater details and the relevant figures moved to the main manuscript. Answer: We moved Figure S4A to the main text. 

Section “Pseudotime Analysis” was added to the Method. “Trajectory inference was conducted using the dynoverse R packages: Set of packages for doing trajectory inference (TI) on single-cell data. https://github.com/dynverse/dyno)23. Briefly, raw count and expression matrices were extracted from the different cell clusters and wrapped into a dynwrap object using dynwrap::wrap_expression. grouping and dimensionality reduction information, these were then added to the dynwrap object using add_grouping and add_dimred respectively. TI methods were selected using the guidelines_shiny tool for our specific dataset. TI methods retained for analysis were slingshot, PAGA-tree and multiple spanning tree (MST). While every method provided specific insight into the data structure upon TI analysis using dynwrap::infer_trajectory, PAGA-tree best recapitulated the structure of other methods combined in our data. Pseudotime visualization was thus obtained using calculate_pseudotime on the PAGA-tree model object. Branches in the cell trajectory represent cells that have alternative gene expression patterns. cHCs were then added to the pseudotime coordinates.”

In figure S4A, I fail to see the ‘dynamic process underlying multi-lineage LPsC differentiation’. Could you provide further explanation?

Answer: We added further explanation in the main text: “Specifically, LPsCs that are devoted to hair cells show development tract from LPsCs to iHC_E14 and iOHC_E16, and finally become more mature OHC at P1 to P7. Other LPsCs specified to supporting cells develop to either IPC or OPC at P1, which mature to DC/PCs at P7.”

 There is virtually no information about how the integration of the OHCs (E14, E16, P1, P7) with the cHCs was performed. This is a crucial step that affects all the data analysis and conclusions moving forward. Step by step details on the workflow should be provided in the methods section.

Answer: We added explanation in the main text: “Beside the standard lognormalization included in Seurat, we tested sctransform’s variance-stabilizing transformation. An important motivation for the development of sctransform was the observation that, even after normalization, the first principal components of various datasets tended to correlate with library size, suggesting an inadequate normalization. For the boxplots in Figure 3C, we calculated the correlation of each feature (gene) with the number of UMIs (the nCount variable). We then placed genes into groups based on their mean expression, and generated boxplots of these correlations. The SCT-normalization generally performs better to adequately normalize genes in the combined dataset, suggesting that technical factors were reduced influence the downstream analysis. The two datasets were then anchored using the top variable genes of each cluster.”

We also added a figure (Figure 3C) to illustrate the outcome of the integration. 

I’m assuming the P33 OHCs were from Yamashita, et al. Why use only those and not the data from other ages from that publication?... it is very confusing to follow which cells were included in which analysis throughout the paper, even when carefully looking at the original publications. This should be more clearly stated throughout the results, figure legends and methods sections. 

Answer: We specified in line 247-248: : “We selected OHCs at E14, E16, P1 and P7 and pooled them with cHCs (cHC1-3) and mature OHC at P33 from Yamashita et al1.”

To make it clearer, we highlighted cells from Yamashita et al1 in figure 3B.

The reason we didn’t use other cells from Yamashita et al1 is that we are interested in the comparison of native HC development and SC-HC conversion. Yamashita et al1 provided us valuable mature OHCs at P33 and the three types of cHCs. Other cell types such as DC/PCs is also present in the other dateset we are using2 (it has native HC development throughout E14, E16, P1 and P7).

I understand you are testing the hypothesis that cHCs may show features of (immature) OHCs. But, why run the integrated analysis only with OHCs? Why not include IHCs to see if they are indeed more distant? And why not run the integration with SCs also? Having common cell types can greatly improve anchoring during integration. Have you considered integrating the full two datasets?

Answer: We have tried including more cell types such as IHCs and SCs. We also tried integrated the whole two datasets. However, the increased cell diversity (hence, more technical batch effect and confounding factors because they are two datasets) will weaken the anchoring effect of cHCs to HCs. OHCs will be anchored to IHCs because they are derived from the same datasets and they have more similarity than with cHCs . 

 Figure 3A shows a UMAP representation of the pooled samples. Without more information of how the two datasets were integrated, the interpretation of this (and all subsequent) plots is weakened. Answer: After modification, Figure 3A was moved to Figure 3B. We also added detailed explanation of how the datasets were integrated. ““Beside the standard lognormalization included in Seurat, we tested sctransform’s variance-stabilizing transformation. An important motivation for the development of sctransform was the observation that, even after normalization, the first principal components of various datasets tended to correlate with library size, suggesting an inadequate normalization. For the boxplots in Figure 3C, we calculated the correlation of each feature (gene) with the number of UMIs (the nCount variable). We then placed genes into groups based on their mean expression, and generated boxplots of these correlations. The SCT-normalization generally performs better to adequately normalize genes in the combined dataset, suggesting that technical factors were reduced influence the downstream analysis. The two datasets were then anchored using the top variable genes of each cluster.”

Lines 220-221: ‘Specifically, a small group of cHC3 are clustered together with OHC_P1…’ it’s unclear from this and from the plot in figure 3A if you are talking about re-clustering after integrating the two datasets, or if you mean that a few cHC3 cells land, on the UMAP plot, on the OHC_P1 region. If the former, how was the clustering done? If the latter, what’s different about those cHC3 cells?

Answer: We could not identify the significant genes differentially expressed in those cells. Therefore, we deleted this sentence to not make any confusion.

Lines 222-227 are confusing and make it even less clear how the data integration was run. Also, were there uncorrected batch effects within the original datasets? Heatmaps of DEGs do not validate UMAPs. How were the top10 DEGs computed? There are hemoglobin and mitochondrial genes in the heatmap, raising concerns that what we are looking at is greatly influenced by batch differences between the two datasets, as opposed to accurately describing the differences and similarities between OHCs and cHCs. At points it seems like the authors confuse clustering with dimensionality reduction/visualisation. Lines 230-233. ‘Heatmap of the top 10 DEGs for each cluster clearly showed that during the dynamic change from cHC1-3, a panel of genes were expressed, and this resembled the natural HC development from E14 to P1 (Figure 3B).’ This is almost discernible on the heatmap (top right corner), but completely overrun by the expression of genes in the bottom right. Again, data integration workflow and batch correction need to be properly addressed and described to support the above statement. Also, to discuss dynamic changes trajectories should be compared, not clusters.

Answer: The explanation of how the data was normalized and integrated were described as stated above.

Why are not P33-OHCs included in figure 3C?

Answer: OHCs_P33 are mature OHC which express SLC26a5. cHCs are close to navie OHC at P1 or P7 at the most extent. That’s why we didn’t included it in this correlation plot. 

 The trajectories in figures S4B-C seem to be more affected by the differences between datasets than by maturation. Again, a clearer explanation on how the datasets were integrated is needed to properly run and interpret this trajectory inference analysis.

Answer: The explanation of how the data was normalized and integrated were described as stated above. Trajectory analysis are more sensitive to the batch effect and cell type difference. It is also expected to see two bifurcation in Figure S4B-C as SC to cHCs conversion are different from nature HC development.

We also modified the Line 325-327 to: “Addition of cHCs to the cell trajectory shows bifurcation branches of cHCs and nature HC development (Figure S3A and B)” as the original statement is inappropriate.

 Line 258. There are 10 genes, not 20, in figure 4A

Answer: The text are corrected. Thank you!

 Figure 4. Atoh1 is overexpressed on cHC3 cells, so it would be expected to show up as such. It should be clarified in the text. 

Answer: We clarified in the main text: “Atoh1 shows as the most significant candidate which is expected because there are ectopic Atoh1 expression in cHCs.”

And further explanation of the interpretation of figures 4A-B would be useful. Is Prox1 significant or not?

Answer: Prox1 was found to be insignificant when using 100 bootstraps and therefore is insignificant when reproduce the data. 

 What’s the rationale behind the umap plot in figure 4C?

Text added: “OHC_P1, OHC_P7 and cHC3 clusters from Figure 3b were subseted out and mapped on UMAP plot (Figure 4C).”

 Lines 307-310. The conclusions stated are not fully supported by the data analysis as it stands. Admittedly, there are some indications that similar expression patterns may be followed by normal HC development and during HC conversion. But, it is not clear from the data analysis how the two datasets were integrated and how the trajectory inference were done. Moreover, to reach conclusions as to the similarity or differences between trajectories, the relevant branches need to be directly compared, an analysis that was not performed here.

 Answer: The explanation of how the data was normalized and integrated were described as stated above.

We modified the conclusion to “Atoh1-mediated SC-to-HC conversion in adult mouse cochleae shares similar patterns with normal SC-to-HC development” in Lines 413-414. We did not use the trajectory shown in SFigure 4B-C as support for this conclusion because the bifurcation in this figure is due to the fact that SC to cHCs conversion are different from nature HC development. We reached this conclusion because of the support from Figure3., which shows cHCs1-3 expressed shared sets of genes with nature HC development and that from Figure 3E, we can see cHC1 and cHC2 is closes to iHC_E14 with r=0.4 and 0.44; while cHC3 is closest to OHC_P1 (r=0.32). 

Also, the direct comparison between SC-cHC conversion is done by comparing their DEG (Figure 3D-E) or by the regulon analysis (Viper analysis in figure 4). 

Line 318. What do the authors mean by “aberrant paths”?

Answer: In line 371, we modified it to: “Moreover, additional genes which are not expressed in nature HC development are also induced during Atoh1-mediated conversion which are likely due to persistent Atoh1 ectopic expression, pervious SC fates, and/or the need for additional transcription factors.”

SPECIFIC POINTS 1. The term ‘master regulator’ appears to be used quite loosely throughout the paper and is hence misleading. For example, in lines 333-336, the authors use this terminology to refer to several differentially expressed transcription factors, whose functions in SC-to-HC fate conversion have not even been studied to date (by their own admission). What makes them ‘master regulators’ as opposed to just ‘transcriptional regulators’? In addition, in lines 92-93, the authors state that master regulators “…are defined as transcription factors that differentially regulate sets of target genes (regulons).” This is not a correct definition of MRs. The authors are describing a regulatory molecule, not a master regulator. Here’s an example of a study that discusses and makes recommendations on this topic: https://www.ncbi.nlm.nih.gov/pmc/articles/PMC3718559/ The authors should instead use ‘regulators’, or ‘transcriptional regulators’ or ‘transcription factors’ where applicable throughout their manuscript.

Answer: This is a real good suggestion. We have made changes to all “master regulator” to either “Transcription factors” or “regulators” to avoid confusion to the readers.

 2. It is well known that distance in the UMAP space doesn’t necessarily indicate transcriptional similarity. The authors even acknowledge this point explicitly in lines 227-229. Yet, they make the claim that cHC3 and OHC_P1 have similar transcriptomes because of cluster proximity in lines 219-222. This should be corrected. The authors are however not wrong in suggesting that cHC3s appear more similar to OHC_P1 than older OHCs; support for this is clearer in the correlation matrix in Figure 3C and the heatmap in Figure 3B.

Answer: We made correction on that point. 

  3. Correction necessary in lines 229-230: A heatmap (of top DEGs) doesn’t validate a UMAP.

Answer: We made correction on that point. 

  4. In lines 237-240, the authors suggest that genes that were highly expressed in cHC1-3 but showed limited expression in OHC_P1 could be downstream targets of Atoh1. This explanation doesn’t make sense: Why would Atoh1-regulated targets be selectively expressed in cHC1-3 when OHC_P1 also express Atoh1?

Answer: We explained in Line 425-427: “Moreover, additional genes which are not expressed in nature HC development are also induced during Atoh1-mediated conversion which are likely due to persistent Atoh1 ectopic expression, pervious SC fates, and/or the need for additional transcription factors.” 

In order to avoid confusion, we modified in Line 320-322: “Genes that were highly expressed in cHC1-3 but had limited expression in wild-type OHC_P1 could be either direct or indirect downstream effectors of persistent ectopic Atoh1 expression, pervious SC fates, and/or the need for additional transcription factors.”

 5. In lines 243-245, the authors suggest that their results in Figures S4B-S4C suggest that maturation stages of cHC1-3 are “similar to the natural HC development”. Those plots do show trajectories across different cell stages/types but cannot be used to make inferences regarding similarity of developmental progression. The heatmap in Figure 3B does provide some support to this statement though, so the authors are advised to reconsider the way they frame this point. 

It is also expected to see two bifurcation in Figure S4B-C as SC to cHCs conversion are different from nature HC development. Therefore, we modified the Line 325-327 to: “Addition of cHCs to the cell trajectory shows bifurcation branches of cHCs and nature HC development (Figure S3A and B)” as the original statement is inappropriate.

 6. Panels in Figure 4 are highly redundant. Most if not all of Figure 4B should be supplemental; anything remaining could be merged with Figure 4A. For example, size of the regulon could be useful in evaluating the data in Figure 4A, so those values could be directly indicated in Figure 4A and the rest of Figure 4B could be moved to a Supplementary Figure.

Answer: We modified the figure 4 according to the reviewer’s suggestion.

 7. It’s unclear how data in Figure 4B, as cited, support the claim made in lines 309-310.

Answer: We corrected in the main text Line 421-422: “Atoh1-mediated SC-to-HC conversion in adult mouse cochleae shares similar patterns with normal SC-to-HC development.”

 8. In line 320, the authors state “These findings further solidify Atoh1-mediated regenerative approaches for hearing restoration…”. It is not clear at all how any of their results allows such a claim.

Answer: We modified the text Line 433-435 to “These findings further suggest that additional transcription factors or regulons need to be manipulated simultaneously or sequentially to ensure the proper and efficient conversion taking place.”

 9. Although the authors have mostly done a decent job of describing the rationale behind the work and their results, there is clearly some room for improvement in writing. This ranges from typos (e.g., line 319), possible mistakes (e.g., “insignificant” in line 266) to unsupported claims (e.g., “…acting as master regulators…” in lines 90-91 where the wording should’ve been “…may act as…”), unintelligible framing (e.g. “…Based on VIPER, the with the top 20 MRs found in the cHC3: OHC_P7 comparative analysis was presented in Figure 4A.” in line 258) and inappropriate content structure (e.g., lines 258-273 contain text written almost like figure legend, not results).

Answer: We improved the writing by correcting those.

---

## [Decision Letter · Decision Letter 1]

21 Feb 2023

PONE-D-22-20296R1Systematic single cell RNA sequencing analysis reveals unique transcriptional regulatory networks of Atoh1-mediated hair cell conversion in adult mouse cochleaePLOS ONE

Dear Dr. Zuo,

Thank you for submitting your manuscript to PLOS ONE. After careful consideration, we feel that it has merit but does not fully meet PLOS ONE’s publication criteria as it currently stands. Therefore, we invite you to submit a revised version of the manuscript that addresses the points raised during the review process.

Thank you for the revised manuscript. As you can see, the two reviewers have divergent views of the revised version. As such, I have combed through the manuscript in detail and identified a few changes that need to be made prior to publication:

1. One of the major concerns of Reviewer 1 and the Editor is the methods, in particular the integration analysis. This can be clarified if all code is easily followed and shared on GitHub. This was also mentioned in initial review:

The data and code sharing should be well annotated and easily followed by someone with a bioinformatics background. In its current state (https://github.com/shutu0302/sc-paper-code), the shared code is not well annotated and appears piecemeal. As inspiration, here's an example of well annotated and shared code that can be easily reproduced (Tosches et al 2018, Science. https://github.molgen.mpg.de/MPIBR/ReptilePallium/). Do include a "readme" file that details where data were obtained (including links), the general outline of the analysis, and with references to each analysis file. Someone should be able to follow this and clearly replicate your results. The main strength of this paper is the analysis methods developed for integration of multiple datasets and therefore this needs to be included.

2. The following needs to be clarified (as included in Reviewer 1's comments):

- Lines 116-118: Figure 1C - add DEG between IHCs vs OHCs. Can be done using scanpy or a wilcoxon rank sum test with the sctransform counts.

- Line 154: reference Figure 2A at the end of this sentence.

- Lines 204-206: Interpretation of Fig 3A. This is not a "bifurcation" per se, but multiple lineages as there are 4 branches. Further, state limitation of this type of analysis as there's one lineage (from LPsC through OHC_P1 and to OHC_P7) that has arrows pointing in both directions (in the middle of OHC_P1), unless that's a separate lineage within the OHC_P1s as the circle dot is not actually on the main lineage. In either case, please clarify language.

- Lines 236-237: Should reference Fig S4A and B, not S3A and B.

- Include a statement in the discussion regarding the difficulty of batch effects and integration of datasets from multiple platforms. 3. Please see comments from Reviewer 2 that are directly on the attached PDF (in blue text).

4. There are a number of grammatical issues that I've identified and detailed below for correction.

Grammar/style:

31: "albeit broad interest" to "albeit of broad interest"

53: take out "the" in front of "the target gene expression"

56: take out "a" in front of "a key to"

70: "regenerating" should be "regeneration"

71: take out "the state-of-the-art"

85: "change" should be "changes"

89: "suggested" should be "suggest"

91: "provided" should be "provides"

91: "of how" not "as how"

108: should read "the Scanpy platform" (insertion of "the")

126-127: remove "the following"

141: should be "Figure S1A"

145: should this be "Supplementary Table 1" or just "Table 1"?

164: should be "Supplementary Table 1"?

173: instead of "invented", how about "developed"

196: change to "The previous study", instead of "previous study"

204: shouldn't this be "the bifurcation of development" instead of "the development"?

222: take out "the" infant of "batch effect"

222-224: combine and modify these two sentences:

"Given that distances between clusters may not be meaningful in

2-d as it uses local notions of distance to construct its high-dimensional graph representation, we also

analyzed the heatmap of the top 10 DEGs for each cluster."

226: take out "the" in front of "natural HC"

228: take out "also" after "We"

232: typo/misspelling "previous" instead of "pervious"

237: change "nature" to "natural"

248: remove "the with"

249: "is" instead of "was"

250: "there are" should be "there is"

250: "represent" instead of "represented"

252: "provide" instead of "provided"

298-299: instead of "the naturally HC development" either "natural HC development" or "the naturally developing HC"

304: "natural" instead of "nature"

315: change to: has been considered a "pro-hair cell gene" (replacing as with "a" and taking out "s")

358: remove "." at the end of single-cell data and replace with "("

647: shouldn't this be "correlated with" instead of "cooperated with"?

Figure 4 legend: should read "Darker values represent higher values" instead of "Darker the color, higher the value"

Note: make all figure references the same throughout the manuscript. An example of inconsistent: "S1 Figure A" (line 141) and "S5 Fig. A" (line 255). Please comb through the manuscript for these.

We look forward to receiving your revised manuscript.

Kind regards,

Taha A Jan, MD

Academic Editor

PLOS ONE

Journal Requirements:

Reviewers' comments:

Reviewer's Responses to Questions

**Comments to the Author**

1. If the authors have adequately addressed your comments raised in a previous round of review and you feel that this manuscript is now acceptable for publication, you may indicate that here to bypass the “Comments to the Author” section, enter your conflict of interest statement in the “Confidential to Editor” section, and submit your "Accept" recommendation.

Reviewer #1: (No Response)

Reviewer #2: (No Response)

2. Is the manuscript technically sound, and do the data support the conclusions?

Reviewer #1: Partly

Reviewer #2: Partly

3. Has the statistical analysis been performed appropriately and rigorously? 

Reviewer #1: No

Reviewer #2: Yes

4. Have the authors made all data underlying the findings in their manuscript fully available?

Reviewer #1: Yes

Reviewer #2: Yes

5. Is the manuscript presented in an intelligible fashion and written in standard English?

Reviewer #1: No

Reviewer #2: No

6. Review Comments to the Author

Reviewer #1: This study conducts timely re-analysis of previously published scRNAseq datasets of the mouse cochlea, attempting to move past the relatively simpler analysis of differential gene expression and focusing on gene networks and searching for ‘master regulator’ transcription factors. As such, the approach is interesting and of use for the community.

Concerning oversights in the data analysis workflow were highlighted in the first revision, with many of these not addressed satisfactorily by the authors.

As one of the stated goals of the paper was to provide an example of integrated analysis of readily available scRNAseq datasets, the authors should provide a lot more detail about how this was achieved.

Moreover, it is still not clear from the data analysis as presented if the authors managed to properly integrate the two datasets to avoid the typical ‘batch effect’ confounding factors.

The authors show some information about data normalisation strategies performed prior to data integration, but they stop short on providing further details.

Also, it is unclear if any data integration/normalisation was performed prior to the trajectory inference analysis.

I still think that there is some indication within the data for a gene expression profile linking cHC3s to P1-OHCs. But this should be re-evaluated under a much stricter and clearer data analysis workflow.

There is a concerning selective usage of subsets of the datasets for specific analysis that is not well justified.

As the manuscript stands, the conclusions stated are not supported by the data analysis performed.

Some specific points follow below. Please know that this is not a comprehensive list of corrections that would need to be made prior to any future publication. Additionally, this list still includes many of the questions raised during the first revision that were not satisfactorily answered, and are not repeated here.

Lines 116-118. You state your cluster analysis can differentiate between IHCs and OHCs. Again, have you tested cluster stability? the plots in figure 1C show DEGs between some cell types and the rest of the dataset. Why not show DEGs between IHCs and OHCs if you can differentiate them?

Figure 2A. the arrows are the bottom are misleading and denote recurrent assumptions made by the authors about the data. The analysis and results displayed in this plot hold no information about directionality of any kind.

Lines 154-156. A reference is made to eigengenes that are actually shown in figure 2A, described in the previous paragraph. Oversights such as this are present throughout the manuscript, making it extremely difficult for the reader to follow what is being analysed and where conclusions are being drawn from.

WGCNA and ARACNE analysis were run on a subset dataset (SC P12, SC P33, cHC1, cHC2 and cHC3), but for SCENIC analysis IHCs and OHCs were also included. Why? If the objective is to perform different analysis strategies that would reveal different aspects of the gene expression profiles of the different subgroups of cells, then surely all analyses must be run on the same datasets. This is even more concerning as the authors aim to compare the outcome of analysis run of different subsets of samples in fig 2D. Moreover, the rationale behind that venn diagram is still not clear.

Additionally, why not include the HCs from P12 in all these analysis?

Figure 3A. The new trajectory inference analysis on the E14-P7 data from Kolla, et al does not identify a split in the trajectory, as the authors state in lines 204-206. The new analysis shows exactly the same pattern as identified in the initial population: LPsCs transition to either DCs or OHCs, without a clear bifurcation.

Lines 206-208. “Specifically, LPsCs that are devoted to hair cells show development tract from LPsCs to iHC_E14 and iOHC_E16, and finally become more mature OHC at P1 to P7. Other LPsCs specified to supporting cells develop to either IPC or OPC at P1, which mature to DC/PCs at P7.”

The trajectory shown in figure 3A does not support these conclusions. For example, there is no clear bifurcation (I count at least 4 branches). There is no clear directionality of the trajectory (there are arrows pointing in both directions within the same branch).

Figure 2C should be in the methods. And, which normalisation method did you use for the data integration?

I still don’t understand why SCs P33 and SCs at P12 from Yamashita, et al, and SCs at P7 from Kolla, et al were not included in the data integration. It is well documented that having cell types in common greatly improves data integration.

Figure 3E… why are correlation values not shown for OHC P33?. I asked this same question in the previous revision… the answer was “OHCs_P33 are mature OHC which express SLC26a5. cHCs are close to navie OHC at P1 or P7 at the most extent. That’s why we didn’t included it in this correlation plot”.

This does not answer my question… and the author’s answer clearly illustrates why OHC-P33 should be in this analysis. If the author’s are correct, and cHCs are indeed more similar to P1 and P7 OHCs, then they should show higher correlation than against OHC P33. If, in contrary, cHCs show higher correlation with OHCs P33 (or SCs), then that could signal that the correlation analysis is confounded by batch effect, that has not been properly corrected.

Lines 236-237: “Addition of cHCs to the cell trajectory shows bifurcation branches of cHCs and nature HC development (Figure S3A and B)”. First, I think you are referring to figures S4A and B. Second, how can you rule out that all the bifurcation is showing you are batch effects?

This new sentence was added in response to a previous question, however was not satisfactorily answered. I copy Q and A:

The trajectories in figures S4B-C seem to be more affected by the differences

between datasets than by maturation. Again, a clearer explanation on how the

datasets were integrated is needed to properly run and interpret this trajectory

inference analysis.

Answer: The explanation of how the data was normalized and integrated were described as

stated above. Trajectory analysis are more sensitive to the batch effect and cell type

difference. It is also expected to see two bifurcation in Figure S4B-C as SC to cHCs

conversion are different from nature HC development.

We also modified the Line 325-327 to: “Addition of cHCs to the cell trajectory shows

bifurcation branches of cHCs and nature HC development (Figure S3A and B)” as the

original statement is inappropriate.

This is extremely confusing. You are saying the SC to cHC conversion is different to natural HC development, yet one of your main conclusions, and the tittle for a whole discussion section, is that SC-to-HC conversion resembles HC development.

The methods section for trajectory inference analysis says that the starting point are raw counts matrices, and there is not mention of a normalisation, data integration, or batch correction method, within the trajectory inference workflow. From this, I can reach two conclusions, either that the TI analysis start from an integrated dataset (then the methods need to be corrected); or that no data integration was performed. If the latter, then the author’s response copied above does not apply to this question and the question still stands, how were the datasets integrated for TI analysis?. Without a clarification, I must maintain my initial observation that the plots in figures S4A-B are showing the batch difference between the two datasets and not a cell differentiation/state bifurcation.

Figure 4. Prox1 should be removed as, as per your own statement, it is not significant.

And this whole analysis requires a lot more clarification. What are the little lines in the middle of the plot? What insight have we gained from this small list of genes?

Other comparisons (OHCs P1 vs P7 and OHCsP1 + cHC3 vs OHCs P7) are shown in fig S5B. why not compare cHC3 vs OHC P1?

The rationale behind figure S5B is still not well explained.

Reviewer #2: (No Response)

7. PLOS authors have the option to publish the peer review history of their article (what does this mean?). If published, this will include your full peer review and any attached files.

Reviewer #1: No

Reviewer #2: No

---

## [Author Response · Author response to Decision Letter 1]

3 Apr 2023

Please see the attachment "Response to Reviewer 1", "Response to Reviewer 2" and "Response to editor". Thank you very much.

---

## [Editor Report · Decision Letter 2]

6 Apr 2023

Systematic single cell RNA sequencing analysis reveals unique transcriptional regulatory networks of Atoh1-mediated hair cell conversion in adult mouse cochleae

PONE-D-22-20296R2

Dear Dr. Zuo,

We’re pleased to inform you that your manuscript has been judged scientifically suitable for publication and will be formally accepted for publication once it meets all outstanding technical requirements.

Kind regards,

Taha A Jan, MD

Academic Editor

PLOS ONE

**Additional Editor Comments (optional):**

**Thank you for the revised manuscript, and in particular for the newly revised source code that will be beneficial to the bioinformatics community.**

**Minor comments:**

I found just a few minor spelling issues that can be corrected during the editorial process - listing them here for your reference:

Line 225 (Discussion): tools not "tool"

Line 231 (Discussion): changes not "change"

Line 340 (Methods): algorithm not "algorism"

**Line 369 (Methods): reverse-engineer not "reverse-engineers"**

---

## [Editor Report · Acceptance letter]

27 Apr 2023

PONE-D-22-20296R2 

Systematic single cell RNA sequencing analysis reveals unique transcriptional regulatory networks of *Atoh1*-mediated hair cell conversion in adult mouse cochleae 

Dear Dr. Zuo:

I'm pleased to inform you that your manuscript has been deemed suitable for publication in PLOS ONE. Congratulations! Your manuscript is now with our production department. 

Kind regards, 

on behalf of

Dr. Taha A Jan 

Academic Editor

PLOS ONE